# Synthesis of New S-Triazine Bishydrazino and Bishydrazido-Based Polymers and Their Application in Flame-Retardant Polypropylene Composites

**DOI:** 10.3390/polym14040784

**Published:** 2022-02-17

**Authors:** Ali Aldalbahi, Bander S. AlOtaibi, Badr M. Thamer, Ayman El-Faham

**Affiliations:** Department of Chemistry, College of Science, King Saud University, P.O. Box 2455, Riyadh 11451, Saudi Arabia; bandr15@hotmail.com (B.S.A.); bader.thamer@yahoo.com (B.M.T.); ayman.elfaham@alexu.edu.eg (A.E.-F.)

**Keywords:** s-triazine, polypropylene, thermal stability, flame retardant, UL-94

## Abstract

In this study six new s-triazine bishydrazino and bishydrazido-based polymers were synthesized via condensation of bishydrazino s-triazine derivatives with terephthaldehyde or via nucleophilic substitution of dichloro-s-triazine derivatives with terephthalic acid hydrazide. The synthesized polymers were characterized by different techniques. The new polymers displayed good thermal behavior with great values in terms of limited oxygen indexed (LOI) 27.50%, 30.12% for polymers **5b**,**c** (bishydrazino-s-triazine based polymers) and 27.23%, 29.86%, 30.85% for polymers **7a**–**c** (bishydrazido-s-triazine based polymers) at 800 °C. Based on the LOI values, these polymers could be classified as flame retardant and self-extinguishing materials. The thermal results also revealed that the type of substituent groups on the triazine core has a considerable impact on their thermal behavior. Accordingly, the prepared polymers were mixed with ammonium polyphosphate (APP) in different proportions to form an intumescent flame-retardant (IFRs) system and were introduced into polypropylene (PP) to improve the flame-retardancy of the composites. The best results were obtained with a mass ratio of APP: **5a**–**c** or **7a**–**c** of 2:1, according to the vertical burning study (UL-94). In addition, the presence of 25% “weight ratio” of IFR in the composite showed great impact and passed UL-94 V-0 and V-1 tests.

## 1. Introduction

Polypropylene (PP) is considered as a vital commodity in the thermoplastic family, and has been extensively employed in numerous diverse usages like transportation, aviation, building and construction materials, the electrical and electronics field, and medical and consumer applications [1,2,3] as a consequence of its chemical and mechanical characteristics, its processability and machinability, cost price ratio, etc. [3]. Hence, raising the fire-retardant activities of plastics material is a critical task for many forms of usage [4,5]. For this reason, employing flame-retardant (FR) material is essential [6,7]. Because of the flammability problem of PP due to its low limiting oxygen index (LOI value is around 17.5%), many scientists have displayed an interest in improving the fire retardancy of polypropylene. They performed intensive work to achieve retardancy of PP. Several researchers recommended low weight material to increase the fire retardancy together with the desired mechanical, thermal and electrical properties. The most popular efficient flame retardant is the one having halogens such as bromine (Br_2_) and antimony trioxide (Sb_2_O_3_) synergistic systems. This type offered a good flame-retardancy system of PP [8]. Implementing this type of flame retardant is limited, as a consequence of releasing a large amount of toxic gases and smoke whenever halogenated flame-retardant burns. An alternative fire-retardant additive is a metal oxide, such as magnesium hydroxide and aluminum hydroxide. However, to acquire the similar grade of halogenated fire retardant, huge amount of metal oxide is needed to annihilate the mechanical properties of polypropylene [9,10,11].

Materials having a heterocyclic moiety such as a s-triazine ring in their backbone chain have acquired special attention as a result of the significant influence on the end-product properties [12,13,14]. The sophisticated structure of the s-triazine ring as well as the exceptional reactivity of the starting material cyanuric chloride (TCT) has been employed as a basic core for preparation of an enormous number of derivatives, as a result of its commercial accessibility and the three chlorine atoms can be exchanged by distinct nucleophiles with the control of temperature that make it valuable in material [15,16,17,18,19,20,21] and industrial applications [22,23,24,25]. Recently, many scientists reported that polymer-based s-triazine has been used in the design and advancement of fire-retardant derivatives [26,27,28], metal adsorption [29], carbon dioxide capture [30,31,32] and covalent organic frameworks [24,25,28]. Also, the design of the structure of s-triazine derivatives could be controlled at different temperatures [33]. These s-triazine derivatives achieved excellent synergistic intumescent flame retardant effect with ammonium polyphosphate (APP) [34,35,36].

Furthermore, attention to s-triazine-hydrazino derivatives has increased [37,38,39] with particular interest in coordination chemistry and supramolecular chemistry [40,41] in addition to its tremendous scope in the area of material chemistry [40,41], complexation with large metal ions [42], and pharmaceuticals [43].

In this regards, and in continuation of our previous work [44,45], we describe herein the synthesis and characterization of two groups of charring agent based on polymers containing s-triazine with hydrazine and hydrazido linkage in their backbone chain. The described derivatives have aniline, p-bromoaniline, p-methoxyaniline, linked to the s-triazine ring. The change of thermal behavior of the prepared compounds based on different substitutes will be studied. Correspondingly, the synthesized charring agent was applied to fire retard PP jointly with APP at different percentages in order to figure out the fire-retardancy characteristics of the described compounds based on the UL-94 V0 rating.

## 2. Materials and Methods

Polypropylene (PP) was obtained from SABIC company (PP 595A, Riyadh, Saudi Arabia). APP was purchased from Hubei Jusheng Technology Co. Ltd (Wuhan, China). All chemicals and solvents were purchased from their commercial sources. FT-IR (cm^−1^) was recorded on spectrophotometer (8201-PC, Shimadzu, Ltd., Kyoto, Japan). Scanning electron microscopy (SEM) was performed on a JEOL (SEM 6380 LA, Tokyo, Japan) to investigate the morphology of the prepared polymers. Differential scanning calorimetry (DSC) was conducted on a DSC, TA instrument Q1000 (New Castle, DE, USA) in the range between 30 °C and 300 °C under N_2_ atmosphere to investigate glass transition (T_g_) of polymers. Thermogravimetric analysis (TGA) data for the prepared polymers were taken by using TA Q500 (New Castle, DE, USA) with a flow rate of 60 mL/min (N_2_ gas), starting from 30 to 800 °C to studied thermal behavior of prepared polymers. The reported analysis for each sample is the average of three reading. 

### 2.1. Synthesis of 2,4-Bishydrazino-6-Substituted S-Triazine Derivatives, ***3a**–**c***

First, compounds **2a**–**c** were prepared as described in Appendix A following the reported methods [41,44]. The NMR spectra agreed with the reported data (Appendix A). 2,4-Bishydrazino-6-substituted s-triazine derivatives were prepared as described in Appendix A following the reported method [41,44] and used directly in the next step.

### 2.2. Synthesis of Polymer-Based S-Triazine Bishydrazino Derivatives, ***5a**–**c***

2,4-Bishydrazino-6-substituted s-triazine derivatives **3a**–**c** (20 mmol) was added portion wise to a solution of terephthaldehyde **4** (2.68 gm, 20 mmol) in 100 mL ethanol containing three drops of acetic acid. The reaction mixture was allowed to cool at room temperature after being refluxed for 24 h. The obtained yellow solid was filtered, washed with cold ethanol, and then dried for 8 h at 60 °C to give the desired polymers **5a**–**c** in yield 80–86%.

### 2.3. Synthesis of Polymer-Based S-Triazine Bishydrazido Derivatives, ***7a**–**c***

First, terephthalic acid hydrazide **6** was prepared according to the reported method [45] and then used for preparation of the desired polymers. Terephthalic acid hydrazide **6** (20 mmol) in 100 mL acetonitrile were added slowly to a solution of 2,4-dichloro-6-substituted s-triazine derivatives **2a**–**c** (20 mmol) in 100 mL acetonitrile in a period of 15 min followed by addition of K_2_CO_3_ (45 mmol) at 25 °C. The reaction mixture was refluxed for 48 h, then solvent was removed under vacuum, and more water was added. The precipitated product was filtered, washed with water, ethanol, and then dried for 8 h at 60 °C to afford the desired polymers **7a**–**c** in yield 80–85%.

### 2.4. UL-94 Test for Flame Retardant of PP Composites

The vertical burning test was examined rendering to the UL-94 test standard with the sample dimension of 130 mm × 13 mm × 3.2 mm. The samples were prepared before compounding. PP and all the additive materials were dried in vacuum oven for 24 h at 70 °C. All tested samples were prepared by mixing PP, APP and **5a**–**c** or **7a**–**c** using a two-roll mill mixing (Rheomixer-XSS-300, Chuang, China) at range of 170 °C–180 °C for 10 min with a roll speed of 100 rpm.

## 3. Results

### 3.1. Synthesis of Polymers

Based on the temperature-dependent reactivity of cyanuric chloride **1**, the first chlorine in **1** was replaced by different aniline derivatives (aniline, p-bromoaniline or p-methoxyaniline) as described in the previous reported method by our group and others [40,41,42,43,44] to afford 2,4-dichloro-6-substituted s-triazine **2a**–**c** (the NMR spectra agreed with the reported ones; Appendix A). The 2nd and 3rd chlorine were replaced by hydrazine group following the method described in literatures [40,41,42,43,44] to give the 2,4-dihydrazino-6-substituted s-triazine **3a**–**c**, which were utilized immediately to react with **4** to offer the target polymers **5a**–**c** in good yields (Figure 1, Table 1). For preparation of the polymers **7a**–**c**, 2,4-dichloro-6-substituted s-triazine 2a–c were reacted with terephthatic acid hydrazide **6** in the presence of K_2_CO_3_ as a base to form the target polymers **7a**–**c** in 80–85% yield as shown in Figure 1.

### 3.2. Characterization of Polymers ***5a**–**c*** and ***7a**–**c***

#### 3.2.1. Solubility

The two class of polymers **5a**–**c** and **7a**–**c** showed weak solubility in different organic solvents, such as 1,4-dioxane, N-methylpyridone, dimethylsulfoxide, dimethylformamide, tetrahydrofuran, acetonitrile, methanol, ethanol, and halogenated solvents, such as CHCl_3_, CCl_4_, CH_2_Cl_2_.

#### 3.2.2. FT-IR Spectrum Analysis

The FT-IR spectrum (Figure 1) for polymers **5a**–**c** showed absorption peaks related to N-H bond in the range 3320 to 3350 cm^−1^. The C-H; sp^3^ and sp^2^ hydrogen have a characteristic peak in the range 2840–2950 cm^−1^. The peaks related to C=N bond of triazine ring observed at 1590 and 1550 cm^−1^. The peaks C=C of phenyl ring were observed at 1600, and 1490 cm^−1^ and the peak at 1270 cm^−1^ was related to the C–N bond of the triazine ring. In addition, the peaks at 757 and 688 cm^−1^ were attributed to the monosubstituted phenyl ring (Table 1). While FT-IR spectra (Figure 2) for polymers **7a**–**c** showed, absorption peaks related to the N–H bond. C–H in the range 3320 to 3330 cm^−1^; sp^3^ and sp^2^ hydrogen has a characteristic peak in the range 2830–2950 cm^−1^. The peaks at 1590 and 1550 cm^−1^ were ascribed to the C=N of the triazine ring. The both peaks at 1630–1620 and 1490–1470 cm^−1^ were ascribed to C=O of the hydrazide group and C=C of the phenyl ring, respectively. The peak at 1270 cm^−1^ was assigned to the C–N bond of the triazine ring (Table 1). The shift in C=N and C–N values was due to the influence of the replacement groups on the aniline ring bonded to the triazine ring. In addition, the elemental analysis for the prepared **5a**–**c** and **7a**–**c** samples agreed with the calculated values as shown in Table 2.

#### 3.2.3. X-ray Diffraction (XRD) Analysis

Figure 3 displays the XRD pattern of the **5a**–**c** polymers; as observed from Figure 3, the spectra of **5c** polymer does not exhibit any sharply peak due to its amorphous nature, which further supports its low compactness. While the two polymers **5a** and **5c** showed a mostly amorphous phase with few crystalline materials. In addition, the 5b polymer have more than five strong fairly sharp diffraction peaks appeared at 2θ = 5–30°, giving a pattern typical of crystalline materials.

Figure 4 displays XRD pattern of the **7a**–**c** polymers, as observed from Figure 4 the polymers showed sharp peaks in the intensity versus scattering angle (2θ) plot. More than six strong fairly sharp diffraction peaks appeared at 2θ = 5–40° for **7c**, **7b** and **7a**, giving a pattern typical of crystalline materials. Based on the XRD spectrum, it was found that the crystallization of **7a**–**c** was higher than that of **5a**–**c**. This might be attributed to the presence of oxygenated functional groups in **7a**–**c**, which make the polymer chains more aligned by hydrogen bonding.

#### 3.2.4. Scanning Electron Microscope (SEM)

The surface morphologies of the polymers **5a**–**c** and **7a**–**c** were examined by using a SEM. The polymer **5a** (Figure 5) showed a globular, multidroplet morphology, and non-uniform porosity distribution with roughness compared to **5b**, while the **5b** showed less porosity and a rougher surface while **5c** (Figure 5) was uniform, smooth with small pores and a rougher surface compared to its analogs **5a** and **5b** (Figure 5). On other hand, the polymers **7a**–**c** (Figure 5) showed more roughness surface with nanosphere morphology. This might be attributed to the different group (CO–NH–NH) rather than the hydrazine (CH=N–NH-)-containing functional groups of the polymers, where in **5a**–**c** there a direct conjugation between the benzene moiety and the functional groups (CH=N–N) which is not the case in **7a**–**c** [46].

#### 3.2.5. Thermogravimetric Analysis (TGA)

TGA was used to investigated the thermal degradation performances of the prepared polymers **5a**–**c**. Degradation curves were obtained for the prepared compounds of the **5a**–**c** (Figure 6 and summarized in Table 3). As observed from Figure 6 and Table 3, the first degradation peaks in the range 50–100 °C for compounds **5a**–**c**, with mass loss about 2–3%, are related to the loss of the solvent content. Compounds **5a**–**c** showed one main degradation peak in the range 320–460 °C, 340–450 °C and 350–450 °C with mass loss 57.88%, 50.50% and 43.70%, respectively. At 800 °C, the residual weight of **5a**, **5b** and **5c** was 16.52, 25 and 31.55%, respectively. This result indicates that the nature of the substituent group in the Para site plays an important role in the thermal behavior of the prepared polymer. Where the electron-releasing group, as OCH_3_ in **5c** and weak electron-attracting group as Br in **5b**, is relatively more thermally stable than the unsubstituted one **5a**.

Table 4 summarizes the thermal properties of the other series (bishydrazido derivatives) **7a**–**c** (Figure 7). As shown in Figure 7 and Table 4 the thermal degradation process of polymers **7a** and **7c** were classified into main three steps; their initial decomposition was between 280–235 °C and 250–330 °C, with mass loss of 10% and 6%, respectively. The second degradation step started in a range of 335 °C to 440 °C and 330 to 430 °C with mass loss of 29% and 23%, for **7a** and **7c**, respectively. The main third degradation step started in the range of 470 °C to 630 °C and 440 to 650 °C with mass loss of 24% and 31%, for **7a** and **7c**, respectively. On the other hand, it was observed that the thermal behavior of the **7b** sample differs from that of samples **7a** and **7c** as it occurred in two main steps. The main first and second degradation steps started in a range of 250 °C to 430 °C and 430 °C to 620 °C with mass loss of 38% and 34%, respectively. This result revealed that polymers-based s-triazine **7a** and **7b** have highly thermal stability compared to their analogs bihydrazino derivatives **5a**,**b**, while **5c** showed higher thermal stability compared to **7c**. The substituent functional groups on the phenyl ring also affected the thermal stability of the prepared polymers. The electron-releasing groups, as OCH_3_ in **7c** and weak electron-attracting group as Br in **7b**, are relatively more thermally stable than the unsubstituted one **7a**. These data agreed with our previously reported data [43].

The formed char could be utilized for the limiting oxygen index (LOI) calculation. To find out the relationship between the limit oxygen index and char residue CR of halogen-free polymer, van Krevelen equation used [47,48,49].
LOI = 17.50 + 0.4*(CR)

The flame retardancy characteristic of any polymer will be enhanced with higher char percentage [46,47]. Materials displaying lower than 26% LOI values are considered flammable, where values more than 26% are classified as self-extinguishing materials [47]. Based on the calculated LOI values as shown in Table 3 and Table 4, the polymer **5a** can be classified as flammable material due to its LOI value less than 26%. In contrast, the polymers **5b**,**c** and **7a**–**c** can be classified as self-extinguishing materials as they have a high LOI value (**5b** = 27.50%, **5c** = 30.12%, **7a** = 27.23%, **7b** = 29.86% and **7c** = 30.85%, respectively) at 800 °C. The results in Table 3 and Table 4 showed that the two **5c** and **7c** have better thermal stability than **5a**,**b** and **7a**,**b**. This indicated that the type of substituent on the phenyl ring attached to the s-triazine core within the polymeric chain, where the methoxy group (electron-donating group) gave the highest char residue and the highest LOI value at 800 °C.

#### 3.2.6. Differential Scanning Calorimetry (DSC)

DSC experiments for the prepared polymers **5a**–**c** with different substituents were evaluated and conducted on a DSC, TA instrument Q1000 thermal analyzer under nitrogen. Scanning from –20 to 300 °C was used to estimate glass transition temperatures (T_g_). The T_g_ value of **5a**, **5b** and **5c** was 88.50, 91.60 and 84.80 °C, respectively (Table 3, Figure 8, and Appendix A).

The thermal properties of **7a**–**c** polymers with different substituent were evaluated and summarized in Table 4, Figure 9 and Appendix A. The T_g_ of the polymers were tested by DSC. The T_g_ values of **7a**–**c** polymers were in the range of 129–150 °C, which were found that T_g_ values of the polymers are slightly similar to each. There are no other obvious exothermic or endothermic peaks in the DSC scans were observed for **5a**–**c** and **7a**–**c** polymers, indicating the high purity sideways with a good stability of polymers.

#### 3.2.7. Flammability of Neat PP and Its Composites UL-94

The achieved polymers **5a**–**c** and **7a**–**c** were mixed with APP to form several intumescent flame retardants (IFRs), which were blended in PP to form PP composites. The UL-94 test was employed to investigate the flame-retardant behavior of pure PP, PP/**5a**–**c**, PP/**7a**–**c** and PP/IFRs composites. The results in Table 5 show that neat PP is highly flammable, and did not pass the UL-94 test. Adding **5c**, **5a** and **5b** at 25% with only 75% PP the composite did not pass the UL-94 rating (Table 5, F1-**5c**, F1-**5a** and F1-**5b**). However, addition of APP to the system made a remarkable improvement in the UL-94 rating of IFR-PP composite. The UL-94 rating was V-2 when the ratio of APP to **5a** was 1:1. The same rating was noticed with **5b** and **5c** (Table 5, F2-**5a**, F2-**5b** and F2-**5c**). In addition, increasing the ratio of APP to **5a** (**5a** 1:2 APP) led to a remarkable improvement in the UL-94 rating and the formed char-layer during combusting, which is unlike what happened in neat PP and other PP composites. For F3-**5a** (Table 5), the UL-94 rating was better and reached the V-1 rating of the UL-94 tests, which is better than PP itself, F1-**5a** and F2-**5a**. As for F3-**5b**, the UL-94 rating also improved and reached the V-1 rating of the UL-94 tests, which is better than PP itself, F1-**5b** and F2-**5b**. As for F3-**5c** (Table 5), the UL-94 rating also improved and reached the V-0 rating of the UL-94 tests, which showed better results compared to pure PP, F1-**5c** and F2-**5c**.

Moreover, F3-**5c** sample reached the UL-94 V-0 rating without dripping, while the other samples showed UL94 V-1 rating. This indicated that F3-**5c** composite has high efficiency in enhancing the flame retardancy behavior of PP amongst all PP-composites. These results established that the mixing of **5a**–**c** with APP in ratio (1:2) improve the flame-retardant behavior of PP. Figure 10 indicated the photographs of F3-**5a**, F3-**5b** and F3-**5c** specimens after UL-94 tests. As indicated in Figure 10, F3-**5a**, F3-**5b**, and F3-**5c** specimens showed char residues, while neat PP showed a melted form without clear char residues.

The **7a**–**c** polymers were mixed with same portion, when the addition of the **7a**, **7b** and **7c** was 25% with only 75% PP, and the composite did not pass the UL-94 rating (F1-**7a**, F1-**7b** and F1-**7c**). However, after APP was added to the system, the IFR-PP composite’s UL-94 rating considerably improved. When the weight ratio of APP to **7a** was 1:1, the UL-94 rating was V-2. The same rating was recorded with **7b** and **7c** (F2-**7a**, F2-**7b** and F2-**7c**). However, increasing the weight ratio of **7a**–**c** (**7a**–**c** 1:2 APP) led to improved UL-94 rating and the formed char layer during combusting was observed compared to the neat PP and other PP composites. For F3-**7a** (Table 5), the UL-94 rating was better than PP itself, F1-**7a** and F2-**7a** and reached the V-1 rating of the UL-94 tests. As for F3-**7a**, the UL-94 rating was better than pure PP, F1-**7b** and F2-**7b** and reached the V-1 rating of the UL-94 tests. As for F3-**7c**, the UL-94 rating also better than pure PP, F1-**7c** and F2-**7c** and reached the V-0 rating of the UL-94 tests. Moreover, the F3-**7c** sample reached the UL-94 V-0 rating without dripping, while the other samples reached the UL-94 V-1 rating. This indicated that F3-**7c** composite is of high efficiency in enhancing the flame retardancy of PP among all PP composites. These results indicated that the combination of **7a**–**c** with APP in weight ratio (1:2) would cause enormous improvement of flame-retardant behavior of PP. Figure 10 indicates photographs of F3-**7a**, F3-**7b** and F3-**7c** specimens after UL-94 tests. It can be seen from Figure 10 that char residues were observed for F3-**7a**, F3-**7b** and F3-**7c**, specimens, while neat PP is in a melted form without any char residues. Also, in comparison with some flame retardants, it seems clear that the prepared polymers can act as a flame retardant in PP/IFR systems as shown in Table 6.

## 4. Conclusions

Two types of polymer-based s-triazine bishydrazino and bishydrazido derivatives (**5a**–**c** and **7a**–**c**) were synthesized, characterized and applied as charring agents together with APP (as acid source) to build an IFR system which could develop the thermal degradation performance of PP. The TGA data revealed that the achieved compounds have shown a good thermal stability and ability of char formation itself. The char residue of **5c**, **5b** and **5a** reached 31.55, 25, and 16.55 wt% at 800 °C with LOI values of 30.12%, 27.50% and 24.11%, respectively; while char residue of **7c**, **7b** and **7a** reached 33.38, 30.90, and 24.32 wt% at 800 °C with LOI values of 30.85%, 29.86% and 27.23%, respectively. Accordingly, replacing of the hydrazine group (C=N–NH-) in the series **5a**–**c** by a hydrazido group (CO–NH–NH) in the series **7a**–**c** increased the thermal stability of the polymers, and in addition the presence of the methoxy in the phenyl moiety in both series exhibited higher thermal stability. Therefore, the prepared derivatives (**5a**–**c** and **7a**–**c**) were blended with APP to make an IFR system. Combustion testing revealed that PP-IFR blends could gain noteworthy LOI values, and pass the UL-94 V-0 rating and the flame-retardant efficacy is developed. When the mass ratio of the components for IFR are 75% APP and 25% **5a**–**c** or **7a**–**c**, the IFR represented the most efficient flame retardancy in PP. The PP-IFR has a UL-94 V2 rating with excessive dripping when the mass ratio of APP and **5a**–**c** or **7a**–**c** is 1:1. This has clearly established that the intumescent flame retardant is very effective in PP. Finally, the substituent effect also affected the thermal stability of the prepared polymers and IFR behaviors as shown from TGA data and UL-94 V0 as well. The electron-releasing group, as methoxy in **5c** and **7c** and weak electron-attracting group as bromine in **5b** and **7b**, are relatively more thermally stable than the unsubstituted one as in **5a** and **7a**. The hydrazido and hyrdazino derivatives with different substituted phenyl ring showed the same influence in the thermal stability and fire retardancy when blended with PP. The mechanism of flame-retardant PP by prepared polymers is still unknown, so more research will be needed in the future.

Finally, the reported polymers herein could be of interest as flame retardant agents and possess special interest for industrial chemistry.

## Data Availability

Data sharing not applicable. No new data were created or analyzed in this study. Data sharing is not applicable to this article.

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
