# Peer review of "Synthesis of New S-Triazine Bishydrazino and Bishydrazido-Based Polymers and Their Application in Flame-Retardant Polypropylene Composites"

_polymers, 2022, doi:10.3390/polym14040784_

Round 1

Reviewer 1 Report

The manuscript entitled “Synthesis of new s-triazine bishydrazino and bishydrazido based polymers and their application in flame retardant polypropylene composites” has improved a lot after revision. But it should address the following comments in detail to meet the requirements to publish on this journal.

  1. The word in the title “flam” was misspelled.
  2. The abbreviations, FTIR, SEM etc, should be illustrated at the first time in the abstract part.
  3. I consist that the LOI value from TGA is no meaning. Delete it or remove it to supporting information part. Because the TGA results
  4. GPC tests should be added to confirm the molecular weight distribution of polymers based s-triazine bishydrazino and bishydrazido derivatives.
  5. Line 82, it should be clarified what atmosphere of the samples are tested by TGA.  
  6. The authors should compare this work with other published work to show the advantages of these new composites.
  7. The authors should cite more newly important references. Like 1016/j.jmst.2021.05.060; 10.1016/j.compscitech.2018.09.024, etc.
  8. Line 163, the authors should explain why these characteristic peaks (C=C, C-N, etc) shift to different values.
  9. Line 204, figure 5, the symbols of these figure should be ordered from a.
  10. These are grammar mistakes in the whole manuscript. Please revise it.

Author Response

Thank you for valuable your comments.

Reviewer 2 Report

The paper from polymers “Synthesis of new s-triazine bishydrazino and bishydrazido 2 based polymers and their application in flam retardant poly- 3 propylene composites’’ is interesting. However there a certain areas which needs to be addressed in order to accept and publish

  1. Any given examples and cited papers for s-triazine have been used in covalent organic framework
  2. In TGA, it is suggested to mention 5 %, 10 %, 50 % and 90 % weight loss temperatures. Which help us to compare with other available data base
  3. In this case, 10% weight loss (T-10%) started at 300 ֯ This is very low comparing with other literature. Any explanation for the early weight loss
  4. TGA for 7 a-c, there is another early decomposition at around 260 ֯ C and the other decompositions are also not consistent. Any explanation for this?
  5. Differential scanning calorimetry (DSC) figures are not at all clear and can not be accepted. They should re-run and present the Tg area clearly and also a full cycle should be presented at least in SI (SI is not available for me now to check)

Author Response

Thank you for valuable comments.

Round 2

Reviewer 1 Report

The authors have addressed the comments. I recomment this manuscript to be published on this journal.